# Hyperspectral Imaging (HSI)—A New Tool to Estimate the Perfusion of Upper Abdominal Organs during Pancreatoduodenectomy

**DOI:** 10.3390/cancers13112846

**Published:** 2021-06-07

**Authors:** Yusef Moulla, Dorina Christin Buchloh, Hannes Köhler, Sebastian Rademacher, Timm Denecke, Hans-Jonas Meyer, Matthias Mehdorn, Undine Gabriele Lange, Robert Sucher, Daniel Seehofer, Boris Jansen-Winkeln, Ines Gockel

**Affiliations:** 1Department of Visceral, Transplant, Thoracic and Vascular Surgery, University Hospital of Leipzig, Liebigstr. 20, D-04103 Leipzig, Germany; dorina.buchloh@medizin.uni-leipzig.de (D.C.B.); sebastian.rademacher@medizin.uni-leipzig.de (S.R.); matthias.mehdorn@medizin.uni-leipzig.de (M.M.); UndineGabriele.Lange@medizin.uni-leipzig.de (U.G.L.); robert.sucher@medizin.uni-leipzig.de (R.S.); daniel.seehofer@medizin.uni-leipzig.de (D.S.); boris.jansen-winkeln@medizin.uni-leipzig.de (B.J.-W.); ines.gockel@medizin.uni-leipzig.de (I.G.); 2Innovation Center Computer Assisted Surgery (ICCAS), University of Leipzig, D-04103 Leipzig, Germany; hannes.koehler@medizin.uni-leipzig.de; 3Department of Diagnostic and Interventional Radiology, University Hospital of Leipzig, D-04103 Leipzig, Germany; timm.denecke@medizin.uni-leipzig.de (T.D.); hans-jonas.meyer@medizin.uni-leipzig.de (H.-J.M.)

**Keywords:** hyperspectral imaging (HSI), pancreatoduodenectomy (PD), celiac artery stenosis (CAS)

## Abstract

**Simple Summary:**

Novel intraoperative imaging systems may have a critical impact on intraoperative decision-making. Hyperspectral imaging (HSI) is one of the leading new imaging systems, providing color pictures of tissue characterization, such as oxygen saturation (StO_2_) and Near-Infrared Perfusion Index (NIR-PI). Several surgical disciplines have already used HSI in detecting tissue perfusion with a proven record of success. To the best of our knowledge, HSI has not been used in the field of pancreatic surgery yet.

**Abstract:**

Hyperspectral imaging (HSI) in abdominal surgery is a new non-invasive tool for the assessment of the perfusion and oxygenation of various tissues and organs. Its benefit in pancreatic surgery is still unknown. The aim of this study was to evaluate the key impact of using HSI during pancreatoduodenectomy (PD). In total, 20 consecutive patients were included. HSI was recorded during surgery as part of a pilot study approved by the local Ethics Committee. Data were collected prospectively with the TIVITA^®^ Tissue System. Intraoperative HS images were recorded before and after gastroduodenal artery (GDA) clamping. We detected four patients with celiac artery stenosis (CAS) caused by a median arcuate ligament (MAL). In two of these patients, a reduction in liver oxygenation (StO_2_) was discovered 15 and 30 min after GDA clamping. The MAL was divided in these patients. HSI showed an improvement of liver StO_2_ after MAL division (from 61% to 73%) in one of these two patients. There was no obvious decrease in liver StO_2_ in the other two patients with CAS. HSI, as a non-invasive procedure, could be helpful in evaluating liver and gastric perfusion during PD, which might assist surgeons in choosing the best surgical approach and in improving patients’ outcomes.

## 1. Introduction

Oxygen delivery and the perfusion of upper abdominal organs mainly arise from the celiac artery (CA) and, to a lesser extent, from the superior mesenteric artery (SMA) [1]. Both arterial territories communicate via the gastroduodenal artery (GDA) and the pancreaticoduodenal arteries [2,3,4,5]. The collaterals in the region of the pancreatic head become essential for the perfusion of these organs in special situations, such as celiac artery stenosis (CAS) [6,7]. 

CAS can be caused by external compression, mainly through the median arcuate ligament (MAL) or internal occlusion through arterial sclerosis [2,8,9]. 

Resection of pancreatic head cancer routinely involves the sacrifice of these communications, which may lead to insufficient perfusion of the upper abdominal organs and sometimes (e.g.,in the case of significant CAS) to their ischemic complications. To prevent these dramatic complications, the GDA clamping test has already been described as a mandatory test to evaluate the hepatic blood flow by ascertaining the satisfactory pulsation of the proper hepatic artery, as well as via Doppler ultrasonography [10,11,12,13]. To the best of our knowledge, there are no quantitative data or well-established algorithms to assess the perfusion of upper abdominal organs and hepatic blood flow during pancreatic surgery before and after GDA clamping.

Hyperspectral imaging (HSI) is a novel medical imaging tool that has proven to be a promising non-invasive, contact-free and non-ionizing method in the analysis of tissue morphology and chemical composition. The camera system is capable of detecting wavelengths within the spectrum of visible light and the near-infrared region (NIR). In brief, this system provides colored pictures for surgeons indicating the oxygen saturation (StO_2_), tissue perfusion (NIR-PI), organ hemoglobin index (OHI) and tissue water index. The application of HSI may have an impact on surgical guidance through tissue characterization [14,15,16]. This study aimed to evaluate the clinical impact of using HSI in determining the perfusion of the liver and the stomach during pancreatoduodenectomy (PD). To our knowledge and according to a recent analysis of the literature, this is the first study to objectively assess the perfusion of upper abdominal organs during PD.

## 2. Materials and Methods 

### 2.1. Study Design

This is a single-institution one-arm prospective exploratory observational study designed in accordance with the declaration of Helsinki. The study was approved by the local Ethics Committee of the Medical Faculty of the University of Leipzig (026/18-ek, 22 February 2018) and registered at Clinicaltrials.gov (accessed on 22 February 2020) (NCT04230603).

### 2.2. Inclusion and Exclusion Criteria

All adult patients undergoing elective open partial or total PD and who had given written informed consent to participate in the study were included. Exclusion criteria were liver cirrhosis and variations of hepatic arterial anatomy (Michels Type III, IV, VI, VII, VIII, IX) [17]. 

### 2.3. Hyperspectral Imaging

For the acquisition of HSI data, the TIVITA^®^ Tissue System (Diaspective Vision GmbH, Am Salzhaff, Germany) was used under standardized conditions as reported previously by our group [18]. HSI measurements did not prolong the regular operative procedure due to its quick applicability (about 10 s per recording and its near-“real-time” possibility of visualization and interpretation). The HSI camera was positioned at a fixed distance of 50 cm above the region of interest (ROI). All ambient lights were switched off to avoid any artifacts during the HSI acquisition. The illumination of the object was performed by six integrated halogen spots that enable spectral data acquisition in the visible and near-infrared spectral range from 500 to 1000 nm with a push-broom HSI camera. Embedded analysis software provided false-color images representing perfusion- and water-related tissue parameters, with an effective number of 640 × 480 pixels. These physiological parameters and their calculations were described by Holmer et al. [19]. Field of view (FOV) and spatial resolution were dependent on the used objective lens and distance. For the presented setup, an 8 × 6.5 cm^2^ FOV and a theoretical spatial resolution of 0.13 mm/pixel were achieved.

### 2.4. Pre- and Intraoperative Assessment 

Contrast-enhanced computed tomography (CT) with arterial and portal-venous scan was performed in all patients with a focus on CAS, vascular anatomic variations, and the possibility of resection. CAS caused by MAL compression was classified as described by Sugae et al. (Table 1) [20].

The indication for surgery was confirmed in our multidisciplinary tumor board in all patients. The presence of intraabdominal metastases and resectability were evaluated immediately after laparotomy. At the time of the lymphadenectomy, the common/proper hepatic artery and the gastroduodenal artery were exposed and controlled. At this time, lactate values and HSI were measured according to our study protocol (Table 2). The GDA was closed off with a bulldog clamp.

Hyperspectral images were acquired intraoperatively with a focus on upper abdominal organs (liver and stomach, Figure 1). The lights in the operation room were switched off for approximately 10 s. After a computation time of 8 s, the analysis software provided false-color images, illustrating the tissue oxygenation (StO_2_), tissue perfusion (Near-Infrared Perfusion Index, NIR-PI), and organ hemoglobin content (OHI) of the liver and stomach. However, before and after GDA clamping, liver and stomach perfusion were examined with the TIVITA^®^ hyperspectral camera system. Larger overview-colored images of the liver and stomach displayed a possible reduction of the perfusion or oxygenation of these organs after GDA clamping (false-colored images show, for example, a color change at the liver surface from red to yellow). After the operation, the hyperspectral data were analyzed with the TIVITA^®^ Suite software to calculate the means of the tissue parameters inside the ROI of the liver and stomach before and after GDA clamping. Furthermore, vital parameters including mean arterial pressure, blood gas analysis, and intraoperative complications were documented in addition to the HSI measurements.

### 2.5. Follow-Up and Endpoints

The primary endpoint was the quantification of the hepatic and gastric perfusion during PD. Secondary endpoints were postoperative ischemic complications, including acute liver failure, stomach ischemia, hepatic abscess, and necrosis of the extrahepatic bile duct. Postoperative ischemic complications were proven either by postoperative CT scan with/without arteriography or by reoperation. Acute liver injury was characterized by markers of liver damage as described by the European Association for the Study of the Liver (EASL) [21,22]. These markers, including transaminases, bilirubin, and prothrombin time (PT), were recorded on the first, third, and fifth postoperative days. 

Postoperative abdominal CT was performed in case of pathological laboratory values to evaluate the morphological and vascular situation of the liver. 

### 2.6. Statistical Analysis 

Obtained categorical data were expressed as absolute and relative frequencies. Descriptive analyses, including mean and standard deviation, were carried out by SPSS 20.0. We used the general linear model with Mauchly’s test of sphericity, as well as boxplot, to analyze and demonstrate the repeated measurements of various laboratory values. A paired *t*-test was used to analyze the tissue parameters determined by repeated HSI measurements with a significance level of *p* < 0.05. 

## 3. Results 

### 3.1. Patient Characteristics 

A total of 20 out of 24 consecutive patients fulfilling the inclusion criteria in 2020 were included in this study. Four other patients were excluded due to various causes: two patients with vascular variation (Michels Type IX), one patient without written consent for the study, and one patient with an accidental lesion of GDA before the test.

The mean age was 65.3 (±8.8 SD) years. There were 9 (45%) women and 11 (55%) men. The mean time between incision and suture was 437.9 ± 110.7 min. Clinical and histological characteristics are demonstrated in Table 3. 

The vascular anatomy could be displayed in all patients through a two-phasic abdominal CT. CAS was detected in four patients (20%) preoperatively: Two patients with Type A, one patient with Type B andone patient with Type C (Figure 2).

### 3.2. Laboratory Tests

There was no postoperative decrease in PT values under 60%. A slight decrease in PT was detected on the first postoperative day only (Figure 3).

Furthermore, there was no critical increase in liver enzymes or bilirubin in the first 3–5 days postoperatively (Figure 4). Consequently, there were no patients with acute liver failure post surgery. Furthermore, none of our patients developed liver abscess, necrosis of the common bile duct, or stomach ischemia during the postoperative course.

### 3.3. Intraoperative HSI-GDA Test

HSI measurements of the upper abdominal organs (liver and stomach) were performed based on the study protocol in all included patients. The *t*-test revealed no significant differences in the StO_2_ (from 57.8% to 55.2%, *p* = 0.38) or OHI (from 67 to 64.5 *p* = 0.65) of the liver before and after GDA clamping in all patients without CAS. The oxygenation of the stomach (StO_2_) in patients without CAS decreased slightly after 30 min of GDA clamping from 88% to 84% (*p* = 0.04) (Figure 5).

Furthermore, we observed an increase in lactate values alongside the decrease in liver oxygenation (StO_2_) 30 min after GDA clamping (Figure 6).

### 3.4. Patients with CAS

According to the preoperative CT used to evaluate the abdominal vascular structure, we detected four patients with CAS caused by external compression through MAL. HSI in patients with CAS Type A revealed no decrease in liver oxygenation 30 min after GDA clamping. On the other hand, we observed a decrease in liver oxygenation after GDA clamping in patients with CAS Type B and Type C (Table 4). Furthermore, HSI data showed a clear improvement of liver StO_2_ after re-opening the GDA (StO_2_ = 64%) and the dissection of MAL (StO_2_ = 74%) in the patient with CAS Type B. A slight decrease in StO_2_ was detected 30 min after GDA re-clamping following MAL release (Figure 7 and Figure 8).

HSI enabled the detection of a remarkable increase in gastric hemoglobin content after GDA clamping in three of these patients (Table 4). The release of MAL was performed in patients with CAS Type B and C only. 

## 4. Discussion

CAS caused by internal stenosis or external compression has been described in 2–7.6% of patients undergoing PD [23]. 

External causes for CAS are mainly compression by MAL, which is a fibrous ligament that passes over the aorta at the level of or superior to the origin of the CA [24,25]. Arteriosclerosis is the main internal cause of CAS in patients undergoing PD. However, in the case of CAS, many major collaterals arise from SMA, such as GDA arcade, ensuring the hepatic arterial blood flow [11,26]. Therefore, dividing GDA arcade during PD may lead to serious ischemic complications postoperatively. However, the clinical impact of the sacrifice of the communication between CA and SMA during PD remains controversial in the surgical community. Pfeiffenberger et al. have already suggested that dividing the GDA and all arterial collateral vessels between SMA and CA during pancreatic head resections with CAS does not influence ischemic complications of upper abdominal organs postoperatively, reporting that only 2 out of 11 patients required MAL division during PD. Regarding the other nine patients without arterial reconstructions, the development of many arterial collaterals from SMA, inferior mesenteric artery and inferior phrenic artery to CA as well as increased portal vein blood flow were suggested to be the reasons for the absence of ischemic complications postoperatively [27]. Song et al. reported another collateral pathway via the dorsal pancreatic artery alongside the common pathway via pancreaticoduodenal arcades, which may replace the right hepatic artery during pancreatic surgery [28]. However, these arterial collateral vessels can be sacrificed during PD. In contrast, other authors have argued that the division of the GDA during PD in patients with CAS without arterial reconstruction leads to severe ischemic complications postoperatively [11,12]. In most cases, CAS is detected preoperatively by a multidetector CT scan with arterial phase (sensitivity: 96%) [11], whereas an intrinsic stenosis can benefit from preoperative arterial stenting. Other kinds of stenosis will be treated, if necessary, during the operation.

Therefore, it is especially important to prove the clinical impact of GDA division during PD in all patients, particularly in patients with CAS. There are, at present, two methods to assess the arterial hepatic blood flow intraoperatively:i.The GDA-clamping test is a routine test performed at the beginning of PD and before GDA division. The weakness of hepatic arterial flow by palpation may reflect the necessity of arterial reconstruction, such as MAL division.

However, this is still a subjective test without any thresholds, which might provide us information to decide for or against arterial reconstruction during PD. Thus, it is of limited value.ii.The use of intraoperative Doppler ultrasonography (US) has already been demonstrated in liver transplantation and later transferred to pancreatic surgery [12,29].

Nara et al. reported that a dramatic decrease in arterial hepatic blood flow during PD after GDA clamping detected by Doppler US is an obvious indication for arterial reconstruction, including the division of the MAL [12]. Gaujoux et al. detected CAS due to MAL compression in 55 (10%) and arteriosclerosis in 62 (11%) patients undergoing PD. Furthermore, about half of them were candidates for arterial reconstruction, including MAL release, due to a significant decrease in arterial hepatic blood flow during the GDA-clamping test via Doppler US. However, they did not define any threshold of this significant decrease in the arterial hepatic blood flow during PD. Furthermore, they only described the improvement of the arterial hepatic blood flow after arterial reconstruction without reaching the values before GDA clamping. The results of the HEPARFLOW study could be useful in this context, but this study is still in progress and recruiting [10]. Normal hepatic artery Doppler waveform, acceleration time (<80 ms), and resistive index (RI; 0.5–0.7) are used in liver transplantation. The deviation of these values may suggest hepatic arterial stenosis in the setting of liver transplantation [30]. To the best of our knowledge, at present there are no definitive criteria or well-established algorithms to evaluate the clinically relevant changes of hepatic arterial blood flow during PD, which might help in making a decision for or against MAL release or any other arterial reconstruction intraoperatively. Additionally, there are no validated thresholds of Doppler US to assess the sufficiency of the hepatic arterial reconstruction during PD. Consequently, none of these tests give sufficient information about the perfusion of upper abdominal organs before and after GDA clamping, which is especially important in patients during PD, because the hepatic arterial and the portal flow may be altered by the reconstruction of the portal vein or the division of arterial collaterals intraoperatively. 

Intraoperative HSI is clearly capable of offering a large variety of information about tissue parameters such as the oxygenation, perfusion, and hemoglobin content of the upper abdominal organs before and after the GDA-clamping test. However, we did not detect obvious changes in these tissue characteristics through HSI in the liver and the stomach or in the laboratory parameters (e.g., lactate values) in most of our patients. There was no need for arterial reconstruction in these patients intraoperatively. Furthermore, there was no acute liver failure in these patients postoperatively. There were four patients with CAS due to external compression. In two patients with CAS Type B and C, we detected an obvious decrease in the StO_2_ of the liver as well as an increase in the OHI of the stomach intraoperatively (Table 4). Thus, a MAL division in these patients was performed. Furthermore, we objectively demonstrated an obvious improvement of tissue oxygenation of the liver and the stomach after MAL division (t4, Figure 7) without any major reduction of liver StO_2_ after GDA re-clamping (t7, Figure 7). This showed the sufficiency of MAL release, and that no further arterial reconstructions were required. We do not have a clear explanation for the increased stomach OHI after GDA clamping. It might be associated with the building of intramural microhematomas or the development of congestion after sacrificing the veins (e.g., right gastric vein) in the region of distal stomach during preparation.

To the best of our knowledge, HSI has not yet been used in the field of pancreatic surgery. We were able to acquire color-coded images of upper abdominal organ perfusion before and after GDA clamping intraoperatively. Furthermore, the improvement of liver perfusion could be detected objectively after MAL division in the patient with CAS Type B (Figure 7). HSI, as a new non-invasive tool, proved highly valuable to objectively assess the perfusion of upper abdominal organs during PD. Our study is still ongoing and recruiting with the aim of reproducing and confirming the current results in a larger cohort.

## 5. Conclusions

Based on this study, the use of HSI during PD to assess the perfusion of upper abdominal organs might help to avoid ischemic complications and to reduce postoperative mortality.

## Figures and Tables

**Figure 1 cancers-13-02846-f001:**
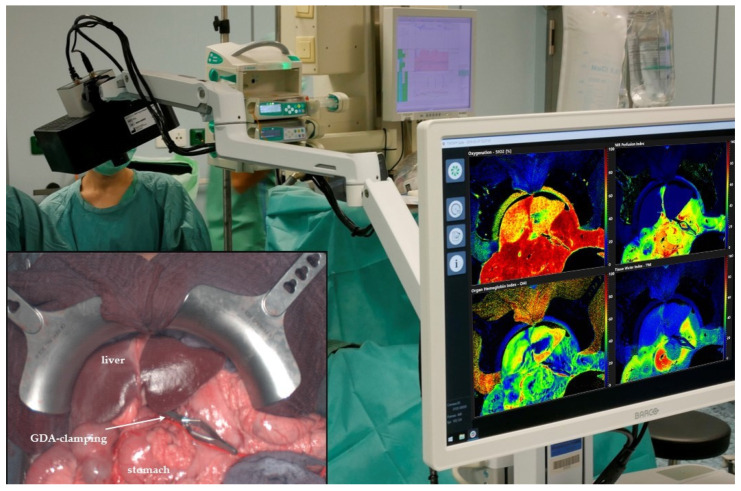
TIVITA^®^ device and HSI camera in the operative setting. Right: The chemical color imaging procedure is calculated with the attached computer. Bottom-left: operative site

**Figure 2 cancers-13-02846-f002:**
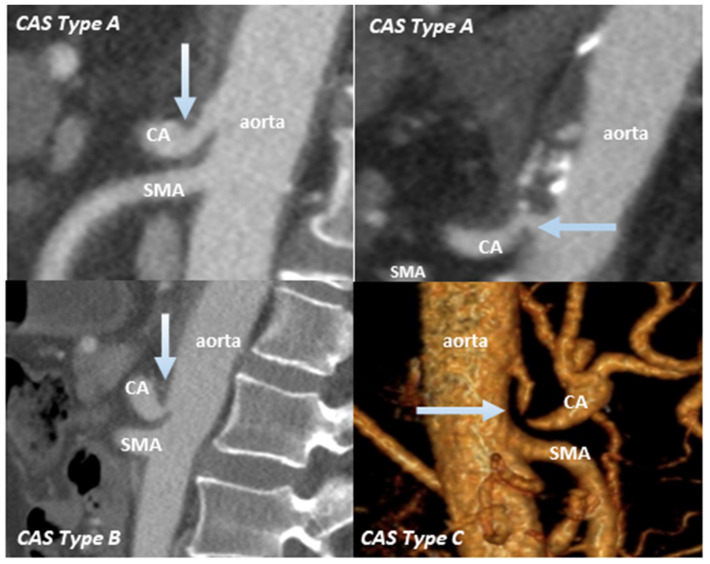
Celiac stenosis in four patients based on preoperative CT images. Arrows show the site of CAS (celiac artery stenosis); CA: celiac artery; SMA: superior mesenteric artery.

**Figure 3 cancers-13-02846-f003:**
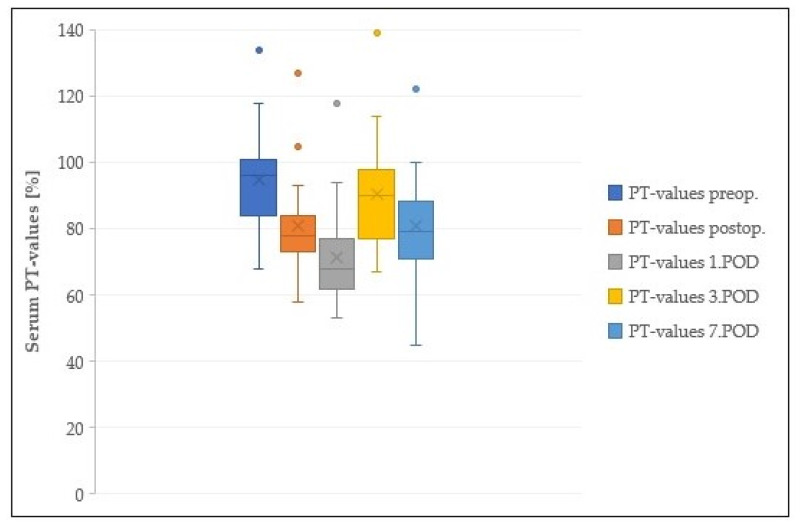
Course of postoperative PT-values; PT: prothrombin time; preop.: preoperative; postop.: postoperative (2 hours after the operation); POD: postoperative day.

**Figure 4 cancers-13-02846-f004:**
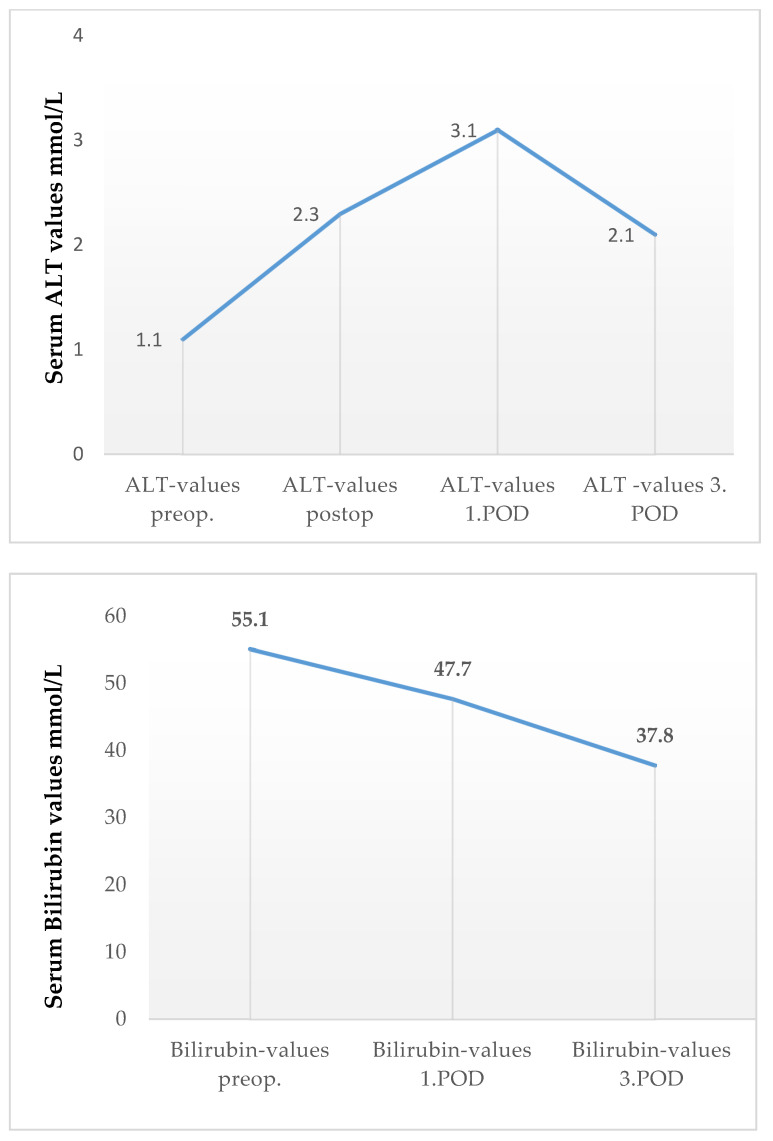
Course of postoperative bilirubin and Alanin Aminotransferase (ALT) values. Entries are means; preop.: preoperative; postop.: postoperative (2 hours after the operation); POD: postoperative day.

**Figure 5 cancers-13-02846-f005:**
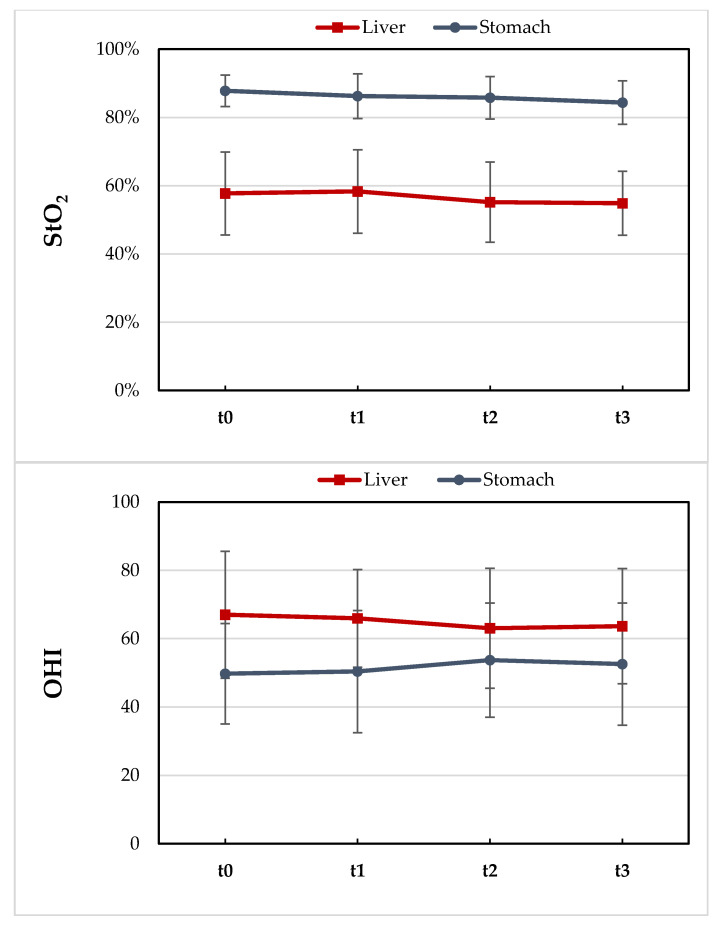
Distribution of tissue oxygenation (StO_2_ in %) and hemoglobin content (OHI) (0–100) in all patients before and after the GDA-clamping test.

**Figure 6 cancers-13-02846-f006:**
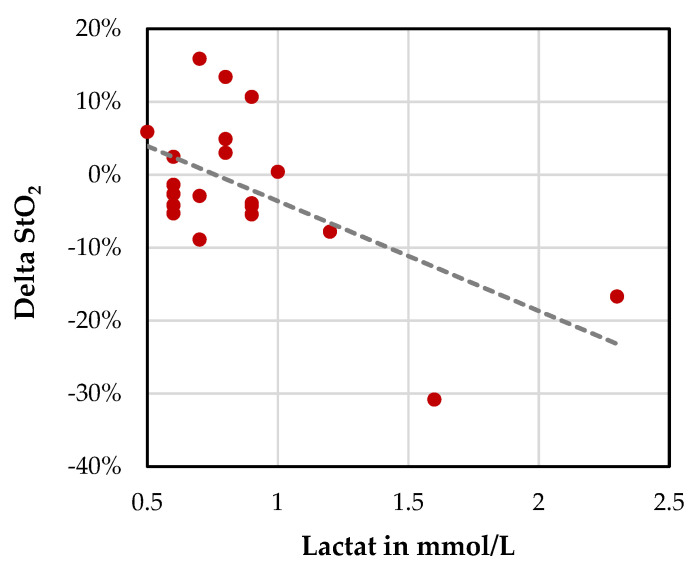
Distribution of lactate values after 30 min and changes of liver oxygenation after GDA clamping (Delta StO_2_ = StO_2 [30 min after clamping]_ − StO_2 [before clamping]_).

**Figure 7 cancers-13-02846-f007:**
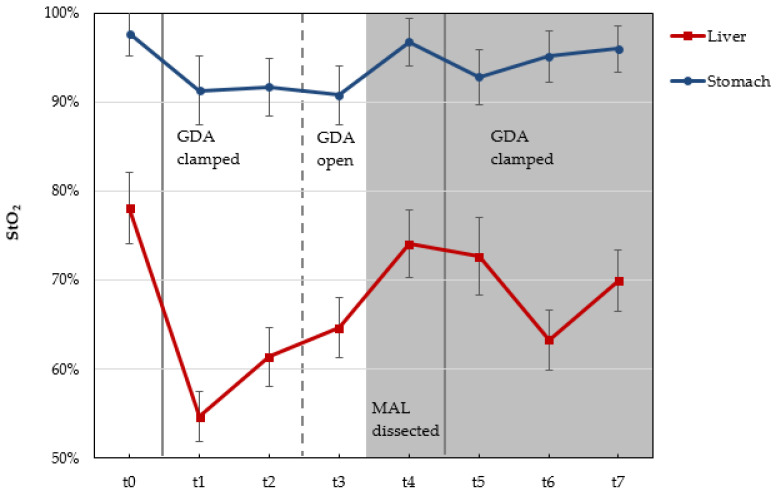
Course of liver oxyegnation (StO_2_ in %) and liver hemoglobin content (OHI) (0–100) in the patient with CAS Type B before and after GDA clamping related to MAL dissection. **t0**: before GDA clamping; **t1**: directly after GDA clamping; **t2**: 15 min after GDA clamping; **t3**: directly after GDA re-opening; **t4**: directly after MAL dissection; **t5**: directly after GDA re-clamping; **t6**: 15 min after GDA re-clamping; **t7**: 30 min after GDA re-clamping. Gray areas show the time after MAL dissection (t4, t5, t6, t7). GDA: gastroduodenal artery; MAL: median arcuate ligament

**Figure 8 cancers-13-02846-f008:**
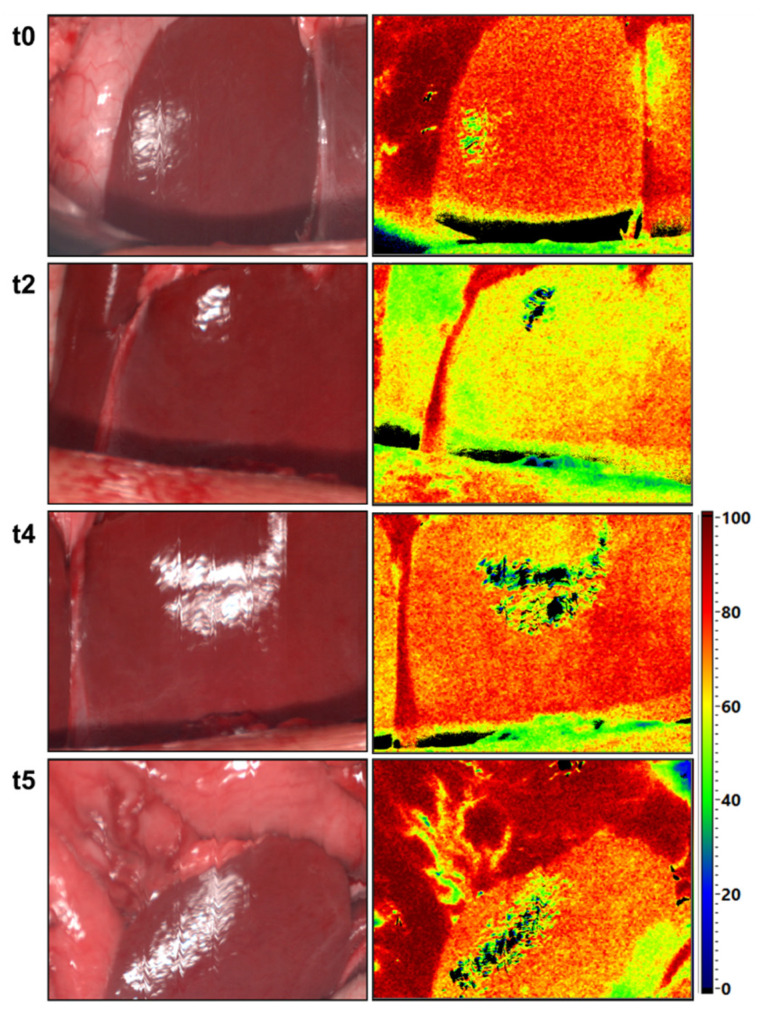
Color images (left) and color-coded images of the tissue oxygenation in % (right) of the liver from patient 3 with CAS Type B. **t0**: before GDA clamping, **t2**: 15 min after GDA clamping, **t4**: directly after MAL dissection, **t5**: directly after MAL dissection and GDA re-clamping.

**Table 1 cancers-13-02846-t001:** Morphologic grading of celiac artery stenosis (CAS) cuased by median arcuate ligament (MAL) compression.

CAS by MAL Compression	Type A	Type B	Type C
Stenosis rate, %	≤50	50–80	≥80
Stenosis length, mm	≤3	3–8	≥8
Distance from aorta, mm	≥5	≥5	small

**Table 2 cancers-13-02846-t002:** Measurement steps (T = time point) according to the study protocol.

Time of GDA Clamping	Selected Measurement
t0 = Before GDA Clamping	HSI + Lactate Measurement
t1 = directly after GDA clamping	HSI
t2 = 15 min after GDA clamping	HSI
t3 = 30 min after GDA clamping	HSI + Lactate Measurement

**Table 3 cancers-13-02846-t003:** Patients’ histological characteristics.

Characteristics	Number of Cases (%)
Entity of pancreatic lesions	
Malignant	14 (70%)
Benign	6 (30%)
Surgical procedure	
PPPD	14 (70%)
Whipple’s procedure	2 (10%)
Total pancreatectomy	4 (20%)
pTNM stage (UICC, 8th Edition)	
IA	2 (10%)
IB	3 (15%)
IIA	2 (10%)
IIB	4 (20%)
III	1 (5%)
IV *	2 (10%)
No malignancy	6 (30%)
pR-classification	
R0	19 (95%)
R1 (pancreas < 1 mm)	1 (5%)
Co-morbidities	
Mild liver disease **	3 (15%)
Diabetes mellitus type II	10 (50%)
Arterial hypertension	10 (50%)
COPD/Asthma	2 (10%)
Auricular fibrillation	3 (15%)
Coronary heart diseases	2 (10%)

* One patient with tumor stage IV had limited hepatic metastasis (<3), which were resected during the surgery. ** There was one patient with liver fibrosis, one patient with advanced hepatic steatosis, and another with hepatitis C. PPPD: pylorus-preserving pancreatoduodenectomy.

**Table 4 cancers-13-02846-t004:** HSI measurements of liver and stomach in patients with CAS.

Patient No.		Before GDA Clamping			30 min after GDA Clamping			
Nr.		StO_2_ in %	OHI (0–100)	Lac in mmol/L	StO_2_ in %	OHI (0–100)	Lac. in mmol/L	Additional Surgical Procedure
**Pat. No. 1** Liver Stomach	Type A	63 89	18 33	0.6	79 83	42 58	0.7	none
**Pat. No. 2** Liver Stomach	Type A	70 91	39 48	0.8	75 94	42 44	0.8	none
**Pat. No. 3 *** Liver Stomach	Type B	78 98	82 35	1.1	61 92	85 79	2.3	dissection of MAL
**Pat. No. 4** Liver Stomach	Type C	67 91	74 60	1.2	59 91	59 70	1.2	dissection of MAL

* HSI values detected after 15 min in this patient; GDA: gastroduodenal artery; StO_2_: tissue oxygenation; OHI: organ hemoglobin index; Lac: lactate; MAL: median arcuate ligament.

## Data Availability

The datasets analyzed during the current study are available from the corresponding author on reasonable request.

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
