# Peer review of "Hyperspectral Imaging (HSI)—A New Tool to Estimate the Perfusion of Upper Abdominal Organs during Pancreatoduodenectomy"

_cancers, 2021, doi:10.3390/cancers13112846_

Round 1
Reviewer 1 Report
The clinical relevance of intraoperative testing the arterial hepatic blood flow and the actually controversial discussed assessment of the perfusion situation has been adequately and plausible described, also the actual lack of an objective measuring method (“subjective test without any thresholds”).
For the described measurement application, Hyperspectral Imaging (HSI) represents an uncomplicated measuring method and offers the potential to provide objective perfusion parameter. The work describes the first application of HSI in this special clinical application area. Certainly, further studies are necessary to obtain a representative data basis and to achieve the goal of determining objective and globally valid assessment and decision parameter.
Detailed comments:
- Fig. 6 is most expressive, the reactions of StO2 for liver and OHI for stomach are significant. An attempt to explain, why for liver only StO2 reacts strongly and not OHI, and conversely for stomach, would be helpful.
- It is described, that there are no significant changes of the HSI parameter for the other patients (without CAS B and C). Is this related to relative changes? How is the variation of the absolute values?
- Other methods (Doppler US) measure an arterial hepatic “blood flow”. This do not include the measurement of StO2, but should correlate with a change of OHI. But for liver HSI-StO2 seems to be the only relevant parameter!
- Is there an additional benefit for the color-coded images (Fig. 7)? Are the perfusion parameters calculated as mean values over the relevant areas in the images?
Author Response
Reviewer # 1 Thanks for the opportunity to review this manuscript. The clinical relevance of intraoperative testing the arterial hepatic blood flow and the actually controversial discussed assessment of the perfusion situation has been adequately and plausible described, also the actual lack of an objective measuring method (“subjective test without any thresholds”).
For the described measurement application, Hyperspectral Imaging (HSI) represents an uncomplicated measuring method and offers the potential to provide objective perfusion parameter.The work describes the first application of HSI in this special clinical application area. Certainly, further studies are necessary to obtain a representative data basis and to achieve the goal of determining objective and globally valid assessment and decision parameter.
1.Fig. 6 is most expressive, the reactions of StO2 for liver and OHI for stomach are significant. An attempt to explain, why for liver only StO2 reacts strongly and not OHI, and conversely for stomach, would be helpful.
We agree with the reviewerand can not fully explain thephenomenonyet.It could be associate with the building of small/ microhematomas intramurallydue tostomach trauma during the preparation h , which might be reflected in anelevated hemoglobin index (OHI). Another explanation could bea congestion development of distal stomach due to sacrifice the veins (e.g. right gastric vein) in this region during the preparation. The reduction of StO2 of the liver is due to reduced arterial blood flow in case of CAS(Page 12)
2.It is described that there are no significant changes of the HSI parameter for the other patients (without CAS B and C). Is this related to relative changes? How is the variation of the absolute values?
Thank you for your comment, StO2 valuesare in % and OHI are dimensionless, the range is [0-100]. We have added the values of changes and the P-values inthetext(Page 8)
3.Other methods (Doppler US) measure an arterial hepatic “bloodflow”. This do not include the measurement of StO2, but should correlate with a change of OHI. But for liver HSI-StO2 seems tobe the only relevant parameter!
Thank youfor your notice, as you have seen in the figures and tables,we couldn ́t detect any relevant changes of the other parameters such as OHI or TWI, it could be explained, that the livergetsother blood supply through the portal system and due to the intrahepatic bile congestion?!
4.Is there an additional benefit for the color-coded images (Fig. 7)? Are the perfusion parameters calculated as mean values over the relevant areas in the images?
Figure 7( corrected to 8)is supposed to illustrate the spatial distribution of the measured tissue oxygenation at selected time points during the procedure. Yes, the mean values over the region of interest are shown in Fig.6 for all time points. We reduced Fig.7 to the exemplary images of one patient(Page 9 &10)
Reviewer 2 Report
The authors present interesting data on the use of HSI to determine the perfusion in the liver and stomach during pancreatoduodenectomy. The data presented are sound, but their presentation and discussion needs to be justified and improved (numerous typos in the manuscript and tables, figures not properly presented, legends to brief, not all abbreviations explained, etc).
My major concern is the use of the TIVITA Tissue system and the parameters (I assume) provided by this system. The authors refer to their reference 6 to look for details on these determined parameters. However, these parameters are not defined appropriately in their own manuscript. So please define StO2, NIR+PI, OHI, TWI, provide the units in which they are presented and explain how they are determined and further used in the analysis (one pixel, or ROI with average and SD)? Looking at the color-coded images in figure 7, the range of the tissue oxygenation within the liver is quite large.
Please note that in "Holmer, A., et al., Hyperspectral imaging in perfusion and wound diagnostics - methods and algorithms for the determination of tissue parameters" these parameters were validated with an precision of ± 20% in the skin. To my opinion the authors have to clarify or discuss the expected accuracy and precision of their measurements in relation to the known literature.
detailed comments:
Abstract:
Aim” to evaluate the key impact of using HSI during pancreatoduodenectomy ” is overrated and not substantiated by the presented method and results.
Results: “Furthermore, HSI data showed a clear improvement of liver StO2 after division of MAL as a modification of surgical procedure. ” I cannot find a substantiated observation in the presented manuscript to claim this result.
Introduction:
Aim: “This study aims to evaluate the effectiveness and efficacy of HSI in determining perfusion of the liver and the stomach during pancreatoduodenectomy (PD)”
How do you define effectiveness? How efficacy? How did you determine those in this study?
Methods:
explain precisely how parameters were measured and analysed, provide information on how "Delta" parameters were determined, make very clear definitions of t0...t8, and how data will be presented.
TWI is mentioned in the manscript, but no data provided, why?
Results:
Add overview image/figure with the important organs/vessels to better understand the procedure and parameters;
Reduce # abbreviations and add a list, check whether used abbreviations are explained (e.g. used in table 3).
Figure 1: indicate dimensions, image presentation (MIP, volume rendering, ..), scale bars, indicate where the CAS is seen, etc
Figure 2: redesign, appropriate horizontal axis and vertical axis (what is course of PT-values??), how defined, units. Check labels, explain them correctly in the figure legend. Errors bars, X, dots indicate what?
Figure 3: No decimal comma’s on vertical axis, explain horizontal axis labels
Text: The oxygenation of the stomach (StO2) decreased significantly after 30 minutes of GDA-clamping (P=0.04) (Figure 4). It is hard to believe this based on the figure data. Decreased compared to what?
Furthermore, we observed an increase of lactate values alongside with the decrease of liver oxygenation (StO2) after 15 and 30 minutes after GDA-clamping (Figure 5). How can I see this relation and provide statistics? How is Delta SO2 defined, is horizontal axis (lactate) one measurement in time (which timepoints?), or also a delta? If the data presented are averages in a ROI, then an SD should be provided. Change decimal comma’s.
Figure 6: decimal comma’s. No error bars? If the data presented are averages in a ROI, then an SD should be provided.
Figure 7: Two patients? Which data sets? Why different time labels? Color bar in %? Why different magnifications?
Table 4: OHI in %, while in the rest of the manuscript is seems to be given as a fraction?
Justify this statement “HSI enabled the detection of a remarkable increase of gastric hemoglobin content after GDA-clamping in the two patients with CAS type B&C (Table 4)” by comparing the measured trends with he expected accuracy and precision, the values measured in other individual patients, etc.
Discussion:
“we detected an obvious decrease of StO2 of the liver and an increase of lactate values intraoperatively.” A decrease compared with what? Where can I see this data?
“we objectively demonstrated an obvious improvement of tissue oxygenation of the liver and stomach after MAL-division without any major reduction after GDA-re-clamping.” Please refer to the data where this is shown.
“This clearly showed the sufficiency of MAL-release and other hepatic arterial reconstructions were not indicated.” What do you mean and based on what observations?
“HSI has been not used in the field of pancreatic surgery yet.” In the introduction this statement was limited to determining the perfusion with HSI. Are you sure that no HSI has been used inter-operatively e.g. for ,argon assessment?
“Furthermore, the effectiveness of MAL- division in patients with CAS could be evaluated intraoperatively for the first time.” How was this effectiveness determined? Where shown?
Conclusions:
“Based on this study, the use of HSI during PD to assess the perfusion of upper abdominal organs might help to avoid ischemic complications and to reduce postoperative mortality. HSI can be applied safely, easily, and sufficiently during pancreatic surgery.“ This conclusion is not clearly based on the results (how to define easy application of HSI?, what is sufficient application?) and not in line with the goals of the manuscript.
Author Response
The authors present interesting data on the use of HSI to determine the perfusion in the liver and stomach during pancreatoduodenectomy. The data presented are sound, but their presentation and discussion needs to be justified and improved (numerous typos in the manuscript and tables, figures not properly presented, legends to brief, not all abbreviations explained, etc).
My major concern is the use of the TIVITA Tissue system and the parameters (I assume) provided by this system. The authors refer to their reference 6 to look for details on these determined parameters. However, these parameters are not defined appropriately in their own manuscript. So please define StO2, NIR+PI, OHI, TWI, provide the units in which they are presented and explain how they are determined and further used in the analysis (one pixel, or ROI with average and SD)? Looking at the color-coded images in figure 7, the range of the tissue oxygenation within the liver is quite large.
Please note that in "Holmer, A., et al., Hyperspectral imaging in perfusion and wound diagnostics - methods and algorithms for the determination of tissue parameters" these parameters were validated with an precision of ± 20% in the skin. To my opinion the authors have to clarify or discuss the expected accuracy and precision of their measurements in relation to the known literature.
- Abstract:
Aim” to evaluate the key impact of using HSI during pancreatoduodenectomy ” is overrated and not substantiated by the presented method and results.
Results: “Furthermore, HSI data showed a clear improvement of liver StO2 after division of MAL as a modification of surgical procedure” I cannot find a substantiated observation in the presented manuscript to claim this result.
Thank you for your comment, this study tries to evaluate the role of using HSI during pancreatoduodenectomy, we need to conduct more patients to get more data about the HSI impact. However, we could detect an improvement of liver StO2 in the patient with CAS typ B after dissection of MAL and GDA-reclamping and no decrease of liver StO2 in other patients. The text has been modified (Page 1)
- Introduction:
Aim: “This study aims to evaluate the effectiveness and efficacy of HSI in determining perfusion of the liver and the stomach during pancreatoduodenectomy (PD)”
How do you define effectiveness? How efficacy? How did you determine those in this study?
The use of HSI helps the surgeon to evaluate the perfusion parameter of abdominal organs such as StO2 and NIR-PI during the surgery. Our group has already published many publications to evaluate the clinical impact of using HSI during abdominal surgery (Refernces 1, 15, 18). We could at least detect no reduction of liver StO2 or NIR-PI in most of our patients after GDA clamping, which could be very supportive alongside intraoperative subjective evaluation to make the right decision regarding the surgical procedure. The Text and the results have been modified appropriately (Page 3, 8 &9)
- Methods:
explain precisely how parameters were measured and analysed, provide information on how "Delta" parameters were determined, make very clear definitions of t0...t8, and how data will be presented.
Thank you for the comments on that. We modified the text, added an explanation of the Delta StO2, The delta StO2 is calculated from the ROI averages of the two timepoints for each patient (not for each pixel). The defintions of t0 to t8 have been clarified in the text (Page 8 & 9)
TWI is mentioned in the manscript, but no data provided, why?
We could not detect any noteable differences in the TWI values before and after the GDA-clamping. It could be explained due to absence of edemas at the time of HSI-Measurement. Therefore, TWI has been removed for the clarity of the manuscript.
- Results:
Add overview image/figure with the important organs/vessels to better understand the procedure and parameters;
Thank you again for your notice, a new figure has been added (Page 4, Figure 1)

Round 2
Reviewer 2 Report
The authors have addressed briefly most of my concerns, but did not incorporate all the information in the materials and methods sections. It was I assume easier for them just to mention them in the legends of the figures.